# Evolution of MALDI-TOF MS Profiles from Lice and Fleas Preserved in Alcohol over Time

**DOI:** 10.3390/insects14100825

**Published:** 2023-10-20

**Authors:** Hanene Benyahia, Philippe Parola, Lionel Almeras

**Affiliations:** 1Aix Marseille Univ, IRD, SSA, AP-HM, VITROME, 13005 Marseille, France; hanenevet@gmail.com (H.B.); philippe.parola@univ-amu.fr (P.P.); 2IHU Méditerranée Infection, 13005 Marseille, France; 3Unité Parasitologie et Entomologie, Département Microbiologie et Maladies Infectieuses, Institut de Recherche Biomédicale des Armées, 13005 Marseille, France

**Keywords:** arthropods, MALDI-TOF MS, duration of ethanol storing, species identification

## Abstract

**Simple Summary:**

Arthropods form an extremely diverse group, including millions of species. Some arthropods, transmitting pathogenic agents, are qualified vectors. To prevent infectious disease outbreaks, accurate identification of arthropods at the species level and distinguishing vectors from non-vectors remain essential. Over the last two decades, matrix-assisted laser desorption/ionization time-of-flight mass spectrometry (MALDI-TOF MS) has emerged as a relevant tool for arthropod identification. Scarce data are available regarding lice and fleas ectoparasites, which present a huge veterinary and human public health problem. These two arthropod species are often collected from livestock in the field and stored in alcohol. However, previous works reported that the duration and conditions of specimen ethanol storage could impede their MS classification. The present study’s aim was to evaluate the performance of the MALDI-TOF MS tool for correct identification of *Pediculus humanus corporis* lice and *Ctenocephalides felis* fleas preserved in alcohol from one to four years. This study highlighted that a correct rate of identification by MS could be obtained for lice and fleas preserved in alcohol for up to four years on the condition that the drying period was sufficiently long enough for accurate identification. The strategy described in the present work could be used as a guideline for MS identification of other arthropod species stored for a long time in ethanol.

**Abstract:**

MALDI-TOF is now considered a relevant tool for the identification of arthropods, including lice and fleas. However, the duration and conditions of storage, such as in ethanol, which is frequently used to preserve these ectoparasites, could impede their classification. The purpose of the present study was to assess the stability of MS profiles from *Pediculus humanus corporis* lice and *Ctenocephalides felis* fleas preserved in alcohol from one to four years and kinetically submitted to MALDI-TOF MS. A total of 469 cephalothoraxes from lice (n = 170) and fleas (n = 299) were tested. The reproducibility of the MS profiles was estimated based on the log score values (LSVs) obtained for query profiles compared to the reference profiles included in the MS database. Only MS spectra from *P. humanus corporis* and *C. felis* stored in alcohol for less than one year were included in the reference MS database. Approximately 75% of MS spectra from lice (75.2%, 94/125) and fleas (74.4%, 122/164) specimens stored in alcohol for 12 to 48 months, queried against the reference MS database, obtained relevant identification. An accurate analysis revealed a significant decrease in the proportion of identification for both species stored for more than 22 months in alcohol. It was hypothesized that incomplete drying was responsible for MS spectra variations. Then, 45 lice and 60 fleas were subjected to longer drying periods from 12 to 24 h. The increase in the drying period improved the proportion of relevant identification for lice (95%) and fleas (80%). This study highlighted that a correct rate of identification by MS could be obtained for lice and fleas preserved in alcohol for up to four years on the condition that the drying period was sufficiently long for accurate identification.

## 1. Introduction

Medical entomology is a discipline that studies arthropods of medical interest affecting animals and humans and more globally impacting public health by causing pests and/or infections [1]. Arthropods form an extremely diverse group, including more than 10 million species, representing nearly 80% of the animal phylum [2]. Some arthropods are qualified as “vectors” by ensuring the transmission of pathogenic microorganisms, such as parasites, bacteria, or viruses, from one vertebrate host to another during their blood meals [3]. These diseases conveyed by arthropods are known as vector-borne diseases (VBDs). To prevent the emergence of VBDs, the monitoring and control of arthropod vectors remains essential. The success of a relevant survey is directly linked to the accurate identification of arthropods at the species level, distinguishing vectors from non-vectors.

Currently, arthropod identification is essentially based on morphological and molecular biology tools [4]. Morphological identification requires the use of dichotomous identification keys in paper or digital formats. However, reliable documentation is not always available for all developmental stages [5]. Moreover, the need for entomological expertise and undamaged specimens are bottlenecks to its widespread use. Conversely, arthropod identification using molecular techniques is accurate, regardless of the specimen’s developmental stage or integrity. Nevertheless, this approach remains time-consuming and expensive, and the availability of target sequences is indispensable [6].

Recently, an alternative tool based on the analysis of protein profiles resulting from matrix-assisted laser desorption/ionization time-of-flight mass spectrometry (MALDI-TOF MS) has been explored for the identification of arthropods [4,7,8]. MALDI-TOF MS has been evaluated as a relatively inexpensive methodology, which is technically reproducible and simple, allowing large-scale and rapid processing for the identification of arthropods, including mosquitoes [9], Culicoides [10], ticks [11], phlebotomine sand flies [12], fleas [13], and lice [14]. Despite the success of this tool for arthropod classification and its broad application to medical entomology [15], some limitations have been reported, as is frequently the case for innovative approaches. Notwithstanding the absence of commercial or sharable reference MS spectra DB, the intra-species reproducibility and inter-species specificity of MS spectra are essential for reliable classification [16]. However, some factors could alter these parameters (i.e., the reproducibility and specificity of MS spectra), such as the specimen compartment selected for MS analysis (same genome, but distinct proteome for different body parts) or the conditions of sample preparation [17]. Optimized procedures and standardized protocols have been established for the identification of mosquitoes, ticks, and fleas by MALDI-TOF MS [17,18,19]. One recent study reported that the cephalothorax with legs from lice appeared as the best compartment for specimen identification by MALDI-TOF MS [14].

The storing conditions of the specimens from field collection until MS analysis and the duration of storage are other factors which could impede arthropod identification. Several studies have reported MS spectra variations between specimens of the same species according to the storing conditions (e.g., frozen vs. in alcohol) [20,21]. Although freezing appeared to be the more efficient preservation method for MS analysis [18,19], it is not systematically possible to refrigerate samples in the field. Then, the storing of specimens in alcohol remains the most frequently used alternative [13,22]. However, this preservation method generally induces modifications of MS profiling from arthropods in comparison with frozen species [20,23]. These MS profile changes could be sufficient to impair matching with its counterpart species from the reference DB stored in another mode [17].

Several studies performed on distinct families of arthropods stored in alcohol underlined the intra-species reproducibility of MS spectra [19,24,25]. The future analysis of specimens stored in alcohol by MALDI-TOF MS becomes possible if new reference MS spectra from counterpart specimens, stored and prepared in the same conditions, are added to the database. Although several studies using the MS profiling tool have demonstrated its success in identifying ticks [20,26] and Culicoides [25,27] stored in alcohol, scarce data are available for lice [16] and fleas [19,21].

Lice and fleas are ectoparasites presenting a huge veterinary public health problem with economic consequences, notably for livestock [28,29]. They have been described as vectors of several pathogens affecting animals and humans. Lice are known as vectors of human diseases, including *Bartonella quintana*, the agent of trench fever; *Borrelia recurrentis*, the agent of louse-borne relapsing fever; *Rickettsia prowazekii*, the agent of epidemic typhus [30,31]; *Yersinia pestis*, the causal agent of plague [32]. Fleas are also vectors of diseases such as bubonic plague, caused by *Yersinia pestis*, and murine typhus, caused by *Rickettsia typhi* [33,34]. Fleas can also transmit *Bartonella henselae*, the agent of cat-scratch disease [35]. These two arthropod species of medical importance are often collected from livestock in the field and stored in alcohol [16,21]. Few studies have evaluated the performance of MALDI-TOF MS for the identification of fleas [13,19] or lice [16] stored in alcohol, and when it was done, specimens were stored in alcohol for a few months without kinetic assessment.

As the duration of specimens stored in alcohol could also influence the reproducibility of MS spectra [36], the present study assessed the stability of MS profiles from lice and fleas stored in alcohol for periods of between one and four years, as well as the efficiency of MALDI-TOF MS for correct identification of these arthropods. For this purpose, one louse (*Pediculus humanus corporis*) and one flea (*Ctenocephalides felis*) species, preserved in alcohol for periods ranging from a few months to four years, were kinetically prepared for MS submission. The reproducibility and stability of the MS profiles over time for each species were evaluated based on the level of significance of the identification accuracy.

## 2. Materials and Methods

### 2.1. Arthropod Rearing and Storage

*Pediculus humanus corporis* lice and *Ctenocephalides felis* fleas were laboratory-reared. Adult *P. humanus corporis* were reared in a climatic chamber (25 °C, relative humidity 80%–90%), as previously described [16,37]. *C. felis* fleas were also reared in a climatic chamber (27 °C, relative humidity 80%). For egg production, *C. felis* were artificially fed with human blood through a parafilm membrane fixed to the plates containing the fleas. The breeding conditions for the complete flea life cycle were carried out as previously described [38,39]. For each louse and flea species, approximately two hundred adult specimens were sedated at −20 °C prior to being stored in 70% (*v/v*) alcohol at room temperature. Lice and fleas were conserved in these conditions until they were subjected to MALDI-TOF MS.

### 2.2. Preparation of Louse and Flea Samples for MALDI-TOF Analysis

Fifteen *C. felis* specimens stored in alcohol were collected monthly for MS analyses from months 6 to 10. Twelve and fifteen specimens per species, *P. humanus corporis* and *C. felis*, respectively, were collected every two months after being stored in alcohol, from 12 until 24 months (i.e., 12, 14, 16, 18, 20, 22, and 24 months) and every six months for those stored in alcohol from 24 until 48 months (i.e., 30, 36, 42, and 48 months). The cephalothorax of each specimen was dissected under a binocular loupe, using forceps and a dissecting blade. Alcohol from cephalothoraxes was evaporated at room temperature (RT) overnight (ON) for approximately 12 h. The cephalothoraxes of the lice and fleas were then homogenized using TissueLyser (Qiagen), with a pinch of glass beads (Sigma, Lyon, France) as a disruptor and 10 µL of homogenization buffer, composed of a 50/50 (*v*/*v*) mix of formic acid (70% *v*/*v*) (Sigma) plus acetonitrile (50% *v*/*v*) (Fluka, Buchs, Switzerland). Cephalothoraxes from two fresh *P. humanus corporis* and *C. felis* were homogenized using the same automated conditions per time point and used as positive controls. Fresh specimens corresponded to laboratory-reared lice and fleas, frozenly euthanized and immediately used. The setting parameters of the TissueLyser for sample homogenization were six cycles of 60 s at a frequency of 30 Hertz. After a quick spin (200 g × 1 min), 1 µL of supernatant from each sample was loaded on the MALDI-TOF MS target plate in quadruplicate (Bruker Daltonics, Wissembourg, France) and covered with 1 µL of matrix solution, as previously described [16,21].

### 2.3. Modification of the Drying Duration of Louse and Flea Samples

Cephalothoraxes from lice and fleas stored in alcohol for 48 months were dried for 12 h (overnight, standard conditions), 18 h, and 24 h at RT. Fifteen *P. humanus corporis* and twenty *C. felis* specimens were tested under each drying condition. The dried samples were then homogenized to be subjected to MS under the same conditions indicated above.

### 2.4. MALDI-TOF MS Parameters and Spectra Analysis

Protein mass profiles were generated using a Microflex LT MALDITOF Mass Spectrometer (Bruker Daltonics, Germany). Details regarding MALDI-TOF MS parameters and MS spectra analysis were identical to previous studies [20,26,40]. Briefly, the reproducibility of the MS spectra was compared visually using four spectra of each sample tested using flexAnalysis v3.3 and ClinPro Tools v2.2 software (Bruker Daltonics). Two representative spectra of each time point were chosen based on the peak position, intensity, and frequency data for each species. The same samples were used for clustering analysis using MALDI-Biotyper v.3.0 software (Bruker Daltonics, Germany). Cluster analyses (MSP dendrogram) were performed to determine how the organisms relate to each other. A fresh specimen of lice and fleas was included in the dendrogram.

### 2.5. Blind Tests

A total of 469 cephalothorax MS spectra from lice (*P. humanus corporis*, n = 170) and fleas (*C. felis*, n = 299), stored in alcohol and collected kinetically until four years, were queried against our homemade DB. For *C. felis*, MS spectra from four specimens that had been stored in alcohol for six months were selected to create reference MS spectra and were incremented in our homemade database using MALDI-Biotyper v.3.0. Software (Bruker Daltonics, Germany). This reference MS spectra DB included MS profiles from several arthropod species (details in Appendix A) and, notably, MS spectra from *P. humanus corporis* lice and *C. felis* fleas species at the adult stage from fresh and frozen specimens as well as those preserved in alcohol for a few months (≤six months) [16,21]. The level of identification significance was determined using the log score values (LSVs) given by the MALDI-Biotyper software v.3.3, corresponding to a matched degree of mass spectra between the query and the reference spectra from the DB. LSVs were obtained for each spectrum of the samples tested in the ranges of the theoretical upper and lower limits of LSVs from 0 to 3, respectively.

### 2.6. Statistical Analysis

Statistical analyses were conducted using GraphPad Prism software 7.0.0 (GraphPad Software, San Diego, CA, USA). After verifying that the LSVs in each group did not assume a Gaussian distribution, nonparametric tests were applied. A comparison of LSVs at different times per species was carried out using Mann–Whitney or Kruskal–Wallis tests when appropriate. All differences were considered significant at *p* < 0.05.

## 3. Results

### 3.1. Assessment of Flea’s Alcohol Preservation Compatibilities with MALDI-TOF MS Analyses

Fifteen specimens of *C. felis* stored in alcohol were submitted monthly (from months 6 to 10) to MALDI-TOF MS analysis. The resulting MS spectra, compared with those from fresh specimens, showed that MS profiles were visually not reproducible between fresh specimens and those stored in alcohol for variable times (Figure 1A). Conversely, the MS spectra from specimens stored in alcohol appeared to be reproducible for each time point but also between time points (Figure 1A). To estimate the reproducibility of these MS spectra, an MSP dendrogram was created using the MS profiles of two specimens per time point. Interestingly, it was observed that MS spectra from fleas stored in alcohol clustered in the same branch, distinct from those of fresh specimens of the same species (Figure 1B). Moreover, the intertwining of the MS spectra from specimens stored in alcohol, independently of the duration of storage, underlined that MS spectra were similar regardless of the length of time they were stored in this buffer.

To assess whether MALDI-TOF MS biotyping could be applied to identify fleas stored in alcohol, MS spectra from 15 specimens stored in alcohol per time point were queried against the homemade MS spectra reference DB upgraded with MS profiles from four *C. felis* stored in alcohol for six months. MS spectra from solely 11 flea cephalothoraxes, stored in alcohol for six months, were then queried against the DB. All (100%) of the 71 specimens analyzed were correctly identified at the species level (Figure 1C). Their LSVs ranged from 1.55 to 2.44. In preliminary studies, an LSV ≥ 1.8 appeared to be a threshold for reliable identification [19]. Here, 78.9% (n = 56/71) of MS spectra succeeded in reaching this threshold (LSVs > 1.8).

### 3.2. Consequences on MS Spectra of Lice and Fleas Storing in Alcohol during Several Years

Despite a delay of 24 h between the last blood meals of these two arthropod species and their collection for sedation at −20 °C prior to being stored in 70% (*v*/*v*) alcohol at room temperature, blood remnant in abdomens was sometimes observed. During the separation of the cephalothorax from the abdomen, blood leaks could, therefore, occur, contaminating the cephalothorax of the specimen. As the aim of this study was to test the stability of cephalothorax MS profiles according to the length of time they were stored in alcohol, all cephalothoraxes contaminated with blood during the dissection step were excluded from the analysis to prevent mismatching of MS spectra due to their corruption by host blood. Consequently, cephalothoraxes from one flea dissected at month 16 and seven lice dissected at months 18 (n = 2), 36 (n = 1), 42 (n = 2) and 48 (n = 2), contaminated with blood remnants, were excluded from the MS submission.

Finally, a total of 289 cephalothorax MS spectra from lice (*P. humanus corporis*, n = 125) and fleas (*C. felis*, n = 164) were subjected to MS analysis. The visual comparison of MS spectra of two specimens per time point, from lice (Figure 2A) and fleas (Figure 2B), indicated the relative stability of profiles for both species, independently of the duration of storage in alcohol from between 12 and 48 months. To estimate the reproducibility of these MS spectra, an MSP dendrogram was performed with MS profiles from a single louse and flea sample per time point. The clustering of MS spectra from lice and fleas onto distinct branches underlined the specificity of these protein profiles (Figure 2C). For both species, fresh specimens were separated from those stored in alcohol, confirming a modification of the profiles according to the storage method. The short distances between the branches of MS spectra from specimens stored in alcohol per species support the high reproducibility of these MS spectra. The absence of ordination according to time in each cluster underlined that the duration of storage in alcohol seems not to affect the MS profiles. Collectively, these data suggest that MS spectra from *P. humanus corporis* and *C. felis* appeared relatively stable throughout the four years of storage in alcohol.

### 3.3. Assessment of Specimen Identification According to Storing Time in Alcohol Based on MS Spectra Query against the MS Reference Spectra Database

MS spectra from the cephalothoraxes of 125 *P. humanus corporis* and 164 *C. felis* stored from one to four years in alcohol were interrogated against the homemade MS reference spectra DB. The query against the database revealed that 97.6% (n = 122/125) of lice and 99.4% (n = 163/164) of fleas were correctly identified at the species level. Their LSVs ranged from 1.34 to 2.37 and 1.19 to 2.39 for *P. humanus corporis* (Figure 3A) and *C. felis*, respectively (Figure 3B). The four misidentified specimens presented an LSV lower than 1.35. As for relevant identification, LSVs should reach 1.8; in that case, 75.2% (n = 94/125) and 74.4% (n = 122/164) of overall lice and fleas, respectively, could be considered accurately classified. Although the proportions of relevant identification were similar for both species, the rate of pertinent classification appeared too low for applying this tool for the identification of lice and fleas stored more than one year in alcohol.

Interestingly, the analysis of the rate of relevant identification per time point and species showed a clear reduction in these proportions after 22 and 24 months of storage in alcohol for lice and fleas, respectively (Appendix A). Effectively, for lice stored during 12 to 22 months, the rate of relevant identification was 87.1% (n = 61/70), whereas from months 24 to 48, this rate fell to 60.0% (n = 33/55), corresponding to a significant decrease (Mann–Whitney test, *p* < 0.001, Appendix A). For fleas, the mean rate of correct identification for months 12 to 20 was 87.8% (n = 65/74), whereas, from months 22 to 48, this rate diminished significantly to 63.3% (n = 57/90; Mann–Whitney test, *p* = 0.042, Appendix A). When the proportion of relevant identification for fleas from months 6 to 10 (78.9%) was taken into account, the global proportion of relevant identification from months 6 to 20 was 83.4%.

### 3.4. Effect of Alcohol Remnants on MS Spectra for Long-Time Storing

At month 16 for lice, only half of the specimens were accurately identified, whereas high proportions of relevant LSVs were obtained before and after this time point (100%) (Additional file S1A). This low rate of relevant identification could likely not be attributed to the storage duration in alcohol. Other factors, such as the modalities of sample preparation, are potentially responsible for this lower reproducibility of MS spectra. To control the quality of the matrix, sample loading, and MALDI-TOF apparatus performance, the matrix solution was loaded in duplicate onto each MALDI-TOF plate, with and without one fresh louse and flea, prepared under the same conditions, which also served as grinding controls, as previously described [9,41]. Here, at all time points, including month 16 for lice, high LSVs (>2.0) with correct identification were obtained for the fresh specimens used as controls of sample preparation. Then, the low LSVs of samples stored in alcohol could, therefore, not be attributed to any buffer or apparatus failures. The only difference between fresh and alcohol-stored lice was the storing mode. In the present study, immediately after dissection, cephalothoraxes from lice and fleas stored in alcohol were dried overnight at RT for at least 12 h. After half a day of drying, no trace of alcohol was visible. As MS spectra were reproducible with intense MS peaks, this drying condition was considered sufficient to evaporate all alcohol traces. However, in a recent study, it was noticed that if arthropod samples stored in alcohol were insufficiently dried, MS spectra of low quality were generated [36].

Then, it was hypothesized that the alcohol trace remaining could occur in these samples, impairing MS spectra matching with the DB. Similarly, it is also possible that for specimens stored more than two years in alcohol, the duration of drying should be increased to eliminate all alcohol traces. To test this hypothesis, 45 lice (15 per time point) and 60 fleas (20 per time point), stored for 48 months in alcohol, were dried between 18 and 24 h and compared to the standard condition (overnight, 12 h). The results of the MS spectra query against the DB indicated an improvement in the accurate identification rate for both species, with the increased duration of drying. The proportion of relevant LSVs reached 80% and 95% for *P. humanus corporis* and *C. felis*, respectively, after 24 h of drying (Appendix A). Although an increase in the mean LSV was noticed for both species with the increase in drying duration, this augmentation was significant only for fleas (*p =* 0.037, Kruskal–Wallis test; Appendix A).

## 4. Discussion

The low cost, speed, and simplicity of MALDI-TOF MS have promoted its widespread application for biotyping microorganisms as well as, more recently, larger organisms such as arthropods [4,15]. Although this innovative approach has shown its potential for the classification of fresh or frozen arthropods, its application to specimens stored in alcohol has required some adaptation of the protocol [20,36]. Alcohol remains one of the most frequent modes used for storing arthropods [21]. It presents the advantage of being cost-effective and simple, and storage at RT for several years is possible [42]. In the present study, the performance of this proteomic tool for the identification of lice and fleas stored in alcohol for several years was tested. As these two ectoparasites engender public and veterinary health problems with economic consequences, notably for livestock [28,29], their monitoring for the distinction of vectors from non-vectors using a rapid, simple, and accurate identification method of the specimen is required.

A previous study analyzed the kinetic MS spectra of cephalothoraxes from *C. felis* stored in alcohol for between one and six months [19]. In this previous study, MS spectra from fleas stored in alcohol were modified in comparison to their fresh counterparts. Nevertheless, a relative stability of the MS spectra was observed for these specimens stored in alcohol. These data suggested the possibility of identifying specimens if counterpart species, stored and prepared under the same conditions, were included in the reference MS spectra DB. Our results support this hypothesis with the correct identification of all the samples, among which nearly 80% were relevantly identified with an LSV > 1.8 [19]. Similar results were obtained for *P. humanus corporis* stored in alcohol from two to twelve months and kinetically analyzed by MS [16]. In both studies, the duration of storing lice and fleas in alcohol prior to MS submission did not exceed one year. As the stability and reproducibility of the species-specific MS spectra from lice and fleas conserved in alcohol through the time during several years were not yet evaluated, it was decided to analyze MS spectra stability of cephalothoraxes from these two ectoparasite species during a period of one to four years.

Interestingly, blood contamination had already been reported during the dissection of engorged hematophagous arthropods, which could alter MS spectra and impair the correct identification of the specimen [12,17,41]. To limit bias in the assessment of MS spectra reproducibility according to the duration of specimens stored in alcohol, the cephalothoraxes from one flea and seven lice contaminated by blood during the dissection step were excluded from the analysis.

For lice, in previous work using MALDI-TOF MS, the proportion of relevant identification for specimens stored in alcohol during the first years reached 93.9% [16], which is consistent with the results obtained in the present work. However, it was observed here that the rate of correct and relevant identification decreased with the longer time of lice preservation in alcohol. Collectively, these results indicate that, for both species, preserving samples in alcohol for more than two years seems to be deleterious for the stability of the MS spectra. Indeed, for lice and fleas, relatively long-term storage in alcohol (more than 22 and 20 months, respectively) could reduce the success of MS identification. This phenomenon affected up to 40% of the sample at 48 months, which is not trivial. However, an acceptable rate of relevant identification could be obtained for specimens stored for nearly two years (20 or 22 months). It is interesting to note that the rate of accurate identification for lice and fleas is lower than for other families of arthropods, such as ticks [23,42], Culicoides [43], and sandflies [44]. Effectively, the storage in alcohol of specimens from these last three arthropod families over the same period (approximately four years) does not appear to dramatically alter their MS profiles. Whereas, for a longer time of storing in alcohol (from 10 to 50 years) [36], standard protocols were shown to be inefficient for tick identification and new methods for sample preparation were required to improve MS spectra. In this previous work, the authors reported that the low quality of MS spectra from tick samples stored in alcohol was attributed to insufficient drying [36].

Based on these data, an optimization of the present protocol was applied to succeed with MS identification of lice and fleas stored in alcohol for four years. An increase in the drying period was then assessed. Our results support that a drying time longer than 12 h (overnight) for louse and flea cephalothoraxes stored in 70% ethanol for more than four years is necessary to evaporate almost all of the alcohol and erase its traces, which alters the MS profile hampering the accuracy of identification. Nevertheless, other factors could induce MS spectra changes, low peak intensities, and/or heterogeneity among replicates [19], such as imperfect dissection, protein degradation during the alcohol storing period, incomplete grinding, or faulty loading on the MALDI target plate [45].

## 5. Conclusions

Low-cost, rapid, and straightforward sample preparation and data analyses were carried out by the MALDI-TOF MS, a competitive tool for arthropod identification. Furthermore, it is now possible to identify relevant lice and fleas stored in alcohol for up to four years. Specimens stored longer than four years in alcohol are required to increase the duration of the drying period to be sure to evaporate all remaining traces of alcohol. In the future, the application of the strategy of increasing the drying time for other arthropod families stored in alcohol for a long time could also be tested to assess whether it could also enhance the proportion and accuracy of specimen identification.

## Figures and Tables

**Figure 1 insects-14-00825-f001:**
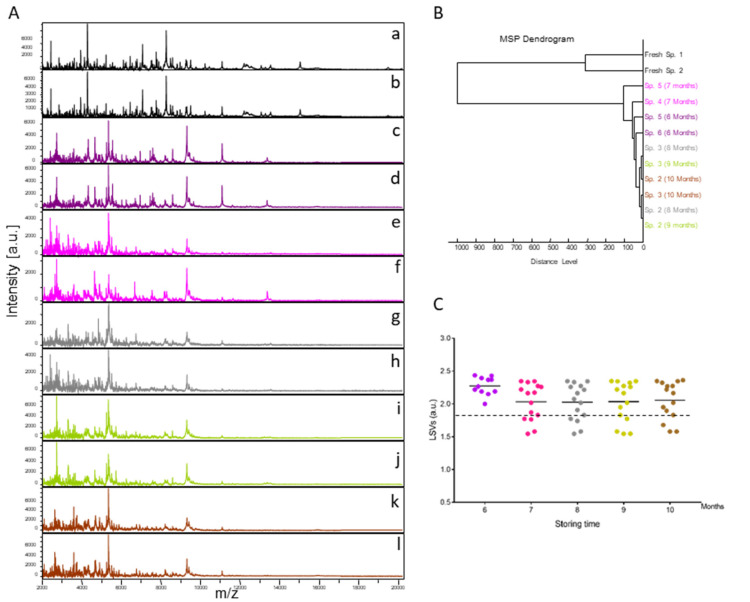
Consequences on cephalothorax MS spectra for their identification of storing fleas in alcohol. (**A**) Representative MS spectra of cephalothoraxes of fresh adult *C. felis* (a, b) and adult *C. felis* stored for 6 (c, d), 7 (e, f), 8 (g, h), 9 (i, j), or 10 (k, l) months in alcohol 70% *v*/*v*. (**B**) Reproducibility and specificity of MALDI-TOF MS spectra from *C. felis* fleas. Two specimens per storage method (fresh vs. alcohol) and duration (from 6 to 10 months in alcohol) were used to construct the MSP dendrogram. The dendrogram was created using Biotyper v3.0 software (Bruker Daltonics, Germany), and distance units correspond to the relative similarity of MS spectra. (**C**) Comparison of LSVs obtained for 15 specimens stored in alcohol tested monthly against the upgraded homemade MS reference database with MS spectra from *C. felis* stored in alcohol. At month 6, as four were included in the DB, only 11 MS spectra were queried against the DB. Dashed lines represent the threshold value for reliable identification (LSV > 1.8). The same color code was used between the different panels for specimens stored in the same conditions. a.u., arbitrary units; LSVs, log score values; *m*/*z*, mass-to-charge ratio.

**Figure 2 insects-14-00825-f002:**
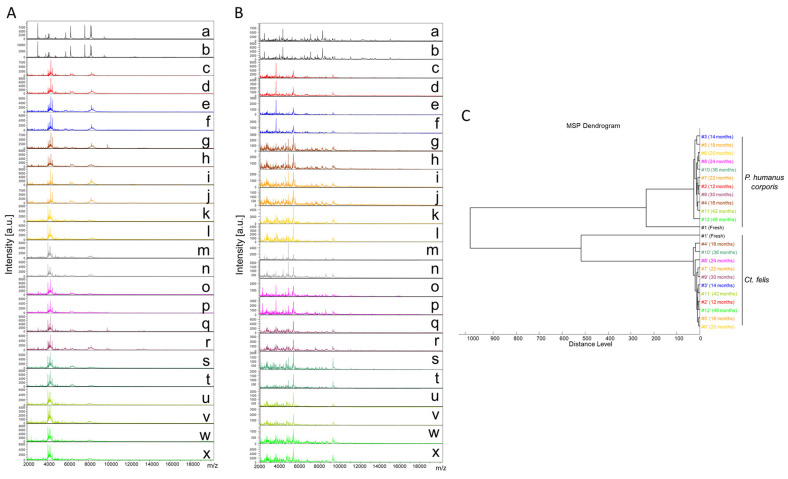
Consequences of alcohol storage time on the stability of cephalothorax MS profiles from lice and fleas. Representative MS spectra of cephalothoraxes of adult *P. humanus corporis* (**A**) and *C. felis* (**B**) from fresh specimens (a, b) or specimens stored in alcohol for 12 (c, d), 14 (e, f), 16 (g, h), 18 (i, j), 20 (k, l), 22 (m, n), 24 (o, p), 30 (q, r), 36 (s, t), 42 (u, v), and 48 (w, x) months. (**C**) Reproducibility and specificity of MALDI-TOF MS spectra from *P. humanus corporis* lice and *C. felis* fleas. One specimen per storage method (fresh vs. alcohol) and length of storage (from 12 to 48 months in alcohol) were used to construct the MSP dendrogram. The same color code was used between the different panels for specimens stored in the same conditions. #1 to #12: specimen number; a.u., arbitrary units; *m*/*z*, mass-to-charge ratio.

**Figure 3 insects-14-00825-f003:**
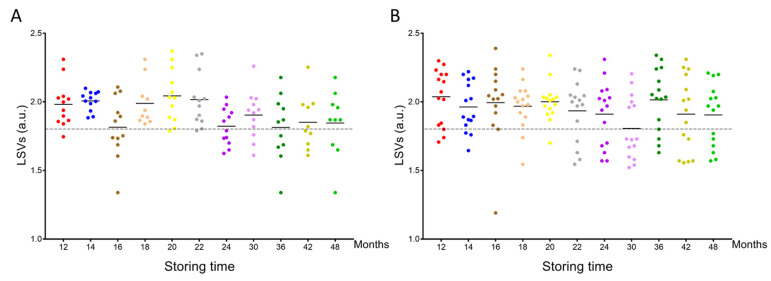
Comparison of LSVs obtained for louse and flea specimens stored in alcohol and kinetically subjected to MS analysis against the upgraded homemade MS reference database. LSVs obtained for cephalothorax MS spectra from *P. humanus corporis* (**A**) and *C. felis* fleas (**B**) stored in alcohol for between 12 months and 48 months were presented. The dashed line represents the threshold value for reliable identification (LSV > 1.8). The same color code was used between the different panels for specimens stored in alcohol for the same length of time. a.u., arbitrary units; LSV, log score value.

## Data Availability

The data presented in this study are available on request from the corresponding author.

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
