# Peer review of "Evolution of MALDI-TOF MS Profiles from Lice and Fleas Preserved in Alcohol over Time"

_insects, 2023, doi:10.3390/insects14100825_

Round 1

Reviewer 1 Report

The work represents an interesting and valuable study devoted to an assessment of MALDI-TOF MS performance for identification of lice and fleas stored in ethanol for several years. The authors recommend to apply extended drying period prior to sample homogenization to obtain the higher ratio of relevant species identifications, especially for the specimens stored longer time. The manuscript is well-written, the methods are properly described, and the data clearly presented. I have only several minor comments.

Minor comments:

L.231-235 As several fleas gave a LSV < 1.8 I could not agree with the statement that “All (100%) specimens were correctly identified”. Although MALDIBiotyper puts the correct species as the first hit, it could not be considered as the correct ID because the given LSV is below the threshold for reliable ID. Please rephrase.

L.234 Looking at the Fig.1C it seems that more than 11 fleas gave LSV below 1.8. Accordingly, please also modify L.305-306.

L.279-282 As mentioned above the specimens with LSV < 1.8 could not be considered as the correctly identified. Please rephrase.

Why was the longer drying period applied to samples stored in alcohol for 48 months only? If possible, could authors add the data for more time points, e.g. 36 or 42 months. Testing of the extended drying for more time points of storage would give the more reliable data to prove the hypothesis that the longer drying might improve the quality of MALDI spectra.

Paragraph 3.4. Is there any evidence that the low identification rate for 16 months is directly related to alcohol remnants? Looking at 14 and 18 months with 100% identifications, it seems that the results for 16 months might be affected by a random or systematic error.

L.394-398, L.408-409 The statements regarding the specimens stored more than 2 years in alcohol are not supported by the results. The data for 48 months are shown only. Please rephrase or add the results supporting these conclusions.

Minor points:

L.145-146 Ref 16 uses overnight drying at 37°C, the present study room temperature. Why is the paper referenced at this point?

L.189-190 Based on the definition, LSV theoretically ranges from 0 to 3. I’m afraid that no LSV equal to 0 or 3 was obtained. Please rephrase.

L.362 from two to twelve

L.364 year instead of years

L.386 less efficient - lower

L.393 to an insufficiently dried - to an insufficient drying

L.394 to succeed with

Reviewer 2 Report

Dear Authors,

The manuscript is well written in a good English. The obtained results represent significant contribution to the science especially from the practical point of view. These finding have also practical use and will help the scientists with the identification of stored samples in alcohol. I do not have a lot of corrections.

Please find attached some specific comments.

L84-86 Why not the whole busy? Why is only cephalothorax giving good results?

L105 Not only veterinary but also are problems in human medicine/public health.

L107 Instead of „for“ it sound better „affecting“.

L128 Never start the sentence with the abbreviation.

L141 Is Ct. felis or C. felis? Please check and correct in the text. In majority of publications is C. felis.

L150 Is fresh= alive or newly euthanized? Or fresh is just put shortly in alcohol?

L161 Ct.felis
